# Epithelial–Mesenchymal Transition by Synergy between Transforming Growth Factor-β and Growth Factors in Cancer Progression

**DOI:** 10.3390/diagnostics12092127

**Published:** 2022-09-01

**Authors:** Masao Saitoh

**Affiliations:** Center for Medical Education and Sciences, Graduate School of Medicine, University of Yamanashi, 1110 Shimokato, Chuo-City, Yamanashi 409-3898, Japan; msaitoh-ind@umin.ac.jp

**Keywords:** EMT, TGF-β, Ras

## Abstract

Epithelial–mesenchymal transition (EMT) plays a crucial role in appropriate embryonic development, as well as wound healing, organ fibrosis, and cancer progression. During cancer progression, EMT is associated with the invasion, metastasis, and generation of circulating tumor cells and cancer stem cells, as well as resistance to chemo- and radiation therapy. EMT is induced by several transcription factors, known as EMT transcription factors (EMT-TFs). In nearly all cases, EMT-TFs appear to be regulated by growth factors or cytokines and extracellular matrix components. Among these factors, transforming growth factor (TGF)-β acts as the key mediator for EMT during physiological and pathological processes. TGF-β can initiate and maintain EMT by activating intracellular/intercellular signaling pathways and transcriptional factors. Recent studies have provided new insights into the molecular mechanisms underlying sustained EMT in aggressive cancer cells, EMT induced by TGF-β, and crosstalk between TGF-β and growth factors.

## 1. Dual Roles of TGF-β in Pathophysiological Conditions

Transforming growth factor-β (TGF-β), a prototypical member of the TGF-β family, binds to transmembrane serine-threonine kinase receptors type Ⅰ (TβR-Ⅰ) and type Ⅱ (TβR-Ⅱ), and transduces canonical signals through Smad proteins [1,2]. TGF-β is the most widely studied cytokine in various types of cells and regulates a broad range of cellular responses, such as proliferation, apoptosis, differentiation, and migration, thereby regulating embryonic patterning, tissue homeostasis, and the immune system. In normal epithelial cells, TGF-β induces apoptosis and suppresses cellular proliferation. The inhibitory growth effects of TGF-β have also been elucidated by several functional experiments in mice [3]. Overexpression of TGF-β reduces the proliferation of keratinocytes and protects mice from carcinogenesis and hyperplasia after the application of chemical carcinogens. In contrast, although genetic alterations in signaling components of TGF-β occur infrequently in various types of cancers, mutations in the *Smad4* gene are observed in approximately 60% and 30% of pancreatic and colorectal cancers, respectively. Genetic mutations and repression of TGF-β receptors and other Smads are also occasionally observed in various cancers. Thus, TGF-β is considered to act as a cancer suppressor in the early stages of cancer (Figure 1).

In the advanced stages of cancer, cancer tissues contain a high amount of TGF-β [4]. TGF-β also regulates cellular traits of various types of stromal cells in the cancer microenvironment, such as immune cells, fibroblasts, and endothelial cells, helping them evade immune surveillance and facilitating proliferation, as well as the motile properties of cancer cells by depositing ECM proteins and promoting lymphangiogenesis and angiogenesis. Consequently, TGF-β promotes cancer progression by modifying the cancer microenvironment. After inoculation of mice with cancer cells, blockade of TGF-β signaling by venous administration of TGF-β inhibitors reduces the number of metastases, but not the growth of primary cancers. In addition, blockade of TGF-β signaling in cancer cells suppresses marked changes in cell morphology such as epithelial–mesenchymal transition (EMT), motile properties, and chemo-resistance of the cells. Therefore, TGF-β in cancer tissues promotes cancer progression by modulating stromal cells in the cancer microenvironment and cancer cells themselves.

## 2. Collective or Single Cancer Cell Migration by EMT

EMT was initially described by developmental biologists, and now is known as a phenotypic conversion that facilitates embryonic development [5]. During EMT processes, epithelial cells lose polarity and their cell–cell junctions, producing migratory mesenchymal cell types in embryogenesis. During the gastrulation stage, the epiblast cells of the primitive ectoderm undergo EMT to differentiate into primary mesenchymal cells. Cells undergoing EMT in embryogenesis are represented by the cadherin switch, termed Type I EMT or complete EMT [6,7]. Conversely, mesenchymal–epithelial transition (MET) is known as the opposite process of EMT [8]. It has been observed that mesenchymal cells of the kidneys undergo MET to differentiate into renal epithelium, resulting in nephrogenesis.

In adults, the EMT program plays crucial roles in pathological and physiological processes, including wound healing, tissue fibrosis, and cancer progression. Keratinocytes adjacent to the wound site start to migrate toward the wound center after they undergo morphological and adhesive changes which resemble those that occur during EMT. In chronic progressive fibrosis of the kidneys and liver in mice, approximately 35% of fibroblasts originated from the epithelial and endothelial cells through EMT, as determined by using a sophisticated mouse model [9]. EMT in wound healing and fibrosis is also known as type II EMT [6].

EMT has also been studied extensively in cancer and is believed to promote chemo-resistance, invasion, and metastasis, and generate circulating tumor cells (CTCs), cancer stem cells (CSCs), and aggressive cancer cells. Surviving CTC is known to express both epithelial and mesenchymal markers, such as E-cadherin and N-cadherin, respectively, and is termed Type III EMT or partial EMT [10]. When CTCs in blood undergo partial EMT, the cells become resistant to anoikis, and extravasate more efficiently to form metastatic colonization. Indeed, it is known that cancer cells with both an epithelial and mesenchymal morphology, but not either alone, show effective metastatic properties [11]. Thus, cancer cells undergoing partial EMT enhance the ECM attachment by achieving mesenchymal characteristics and exhibit collective cell migration through their remaining epithelial characteristics [12]. However, the possibility that invading cancer cells have undergone complete EMT cannot be ruled out because it may not be possible to detect complete EMT among invading cells in vivo.

Interestingly, a heterophilic cell–cell adhesion between E-cadherin on the cancer cell membrane and N-cadherin on the membrane of fibroblasts or cancer-associated fibroblasts (CAFs) enables fibroblast-led collective cancer cell migration [13,14]. More importantly, it is likely that the interaction between cancer cells and fibroblasts through a heterophilic cell–cell adhesion is bridged by cancer cells undergoing partial EMT through a hemophilic cell–cell adhesion. Therefore, heterotypic interactions between non-EMT cancer cells and either EMT cells or fibroblasts undergo collective cell migration to promote metastasis during cancer progression (Figure 2).

Collective cancer cell migration exhibits multicellular clusters resembling a bunch of grapes and depends on cell–cell adhesion through epithelial adhesion molecules of solid cancer cells [15]. On the other hand, single cancer cell migration depends on integrin-mediated interactions between cancer cells with a mesenchymal trait and ECM. In pancreatic cancer, active *K-RAS* mutations are very frequently detected. During the early stages of carcinogenesis, active RAS signals result in the sustained activation of MAPK pathways, which reportedly help overcome the cancer suppressing effects of TGF-β by promoting Smad degradation (Figure 1) [16]. When active *K-RAS* mutations are introduced in normal epithelial cells which had been sensitive to growth inhibition by TGF-β, they confer resistance to TGF-β (Figure 1). Thus, these normal epithelial cells with active *K-RAS* mutations proliferate more rapidly by suppressing TGF-β signals, and become exposed to a large amount of TGF-β autonomously secreted from the cells themselves. Consequently, cancer cells in which signaling molecules of TGF-β are not mutated showed cooperation between RAS signals and TGF-β signals to induce drastic EMT (see below) and efficient invasion into stromal tissues, possibly leading to single cancer cell migration.

## 3. EMT Transcription Factors (EMT-TFs)

It is widely known that both “complete EMT” and “partial EMT” are induced by several common transcription factors, referred to as EMT-TFs. Representative EMT-TFs include the basic helix–loop–helix (bHLH) factor TWIST (TWIST1 and TWIST2), the SNAIL family of zinc finger transcription factors (SNAIL, SLUG, and SMUC), and the two-handed zinc finger factors of the zinc finger E–box-binding homeobox (ZEB) family proteins (ZEB 1/δ-crystallin/E-box factor [δEF1] and ZEB2/Smad-interacting protein [SIP]1) [10].

TWIST was originally identified in *Drosophila* as a zygotic gene required for dorsoventral patterning, and its gene mutation is reported to cause Saethre Chotzen Syndrome, an autosomal dominant human craniosynostosis syndrome [17]. *Twist1* null mice die at approximately E11 due to severe phenotypic abnormalities including neural tube closure defects. *Twist2* null mice die immediately after birth due to severe inflammation and cachexia. TWIST consists of a nuclear localization signal and two glycine-rich domains in the N-terminal region, a bHLH domain in the central region, and a WR (tryptophan and arginine) motif, also known as a Twist box, in the C-terminal region [17,18]. The Twist box is required for prostate cancer cells to colonize metastatic lung lesions and extrathoracic metastases [19], while the glycine-rich domains, which are absent in Twist2, may be used to interact with proteins, resulting in differences in protein function between TWIST1 and TWIST2 [20]. Phosphorylation of TWIST1 regulates its transcriptional activity and protein stability. PKA phosphorylates Thr125 and Ser127 of TWIST1, and then enhances its dimerization and DNA binding through its bHLH domain [21]. Although Ser68 phosphorylation by ERK increases the stability of TWIST1, the small C-terminal domain phosphatase 1 (SCP1) dephosphorylates Ser68 and promotes degradation [22,23]. Phosphorylation at Ser42/Thr121/Ser123 by AKT, Ser144 by PKCα, and Ser18/Ser20 by casein kinase 2 regulates protein stability through modifying ubiquitin-dependent degradation [24,25,26,27]. Other post-translational modifications, including ubiquitin, acetylation, and methylation, are also reported to regulate TWIST function during cancer progression.

SNAIL is widely known to be the most important transcription factor for EMT [28], because *Snail* gene knockout mice exhibit severe gastrulation defects, resulting in embryonic lethality at an early stage of development. The SNAIL family members consist of a SNAG domain in the N-terminal region, a nuclear export sequence and a serine-rich domain (SRD) in the central region, and several zinc finger clusters in the C-terminal region [29,30,31]. SNAG is essential for binding to other transcriptional co-regulators, and SRD is needed to regulate protein stability dependent on phosphorylation. SLUG contains another unique “SLUG” domain with unknown functions, whereas SMUC does not possess both SLUG and SRD [20,32]. SNAIL is phosphorylated at Ser104/Ser107 and Ser96/Ser100 by casein kinase 1 and glycogen synthase kinase 3β (GSK3β), respectively [33,34]. These phosphorylation events confer interaction with E3-ligase proteins, such as β-TrCP, and subsequent ubiquitin-dependent degradation. In addition, phosphorylation of Ser246 by p21-activating kinase 1 (p21PAK1) and Ser82/Ser104 by ERK causes nuclear localization and accumulation, whereas that of Ser11 by protein kinase D1 (PKD1) causes nuclear cytoplasmic translocation to inhibit the transcriptional activity of SNAIL [35,36]. Furthermore, the function of SNAIL is regulated by ubiquitination, acetylation, and glycosylation [20].

Similar to SNAIL, SLUG is also phosphorylated at multiple sites: Ser87 by ERK, Ser92/Ser96/Ser100/Ser104 by GSK3β, and Ser158/Ser254 by p21PAK4 [37,38,39]. These phosphorylation events and several post-translational modifications, including ubiquitination, sumolylation, and acetylation, control functions of SLUG. In contrast to SNAIL and SLUG, the roles and post-translational modifications of SMUC have not yet been elucidated. *Smuc* null mice show almost normal phenotypes [40]. Thus, SMUC seems to not be essential for normal development in mice.

The ZEB family members, ZEB1 and ZEB2 (ZEB1/2), are also known to be crucial factors for EMT, because knockout phenotypes of *Zeb1* and *Zeb2* show postnatal death with severe defects in hematopoiesis and skeletal bones, and embryonic lethality with severe defects of cranial neural crest cells, respectively. They consist of several zinc finger clusters at both the N- and C-terminal regions. Each of these two clusters located in the N- and C-terminal regions binds to the paced E-boxes in the E-cadherin promoter, and then represses the expression of E-cadherin. ZEB1/2 also directly bind to the E-box in many gene promoters through the zinc finger clusters. In the central region, ZEB1/2 have a Smad-binding domain (SBD), C-terminal binding protein (CtBP) domain, and homeodomain. The homeodomain of ZEB1/2 regulates transcriptional control by modifying the protein–protein interaction. The SBDs in ZEB1 and ZEB2 are approximately 40% similar in amino acid sequence, but have dramatically low structural similarity [41]. In response to TGF-β, activated Smads bind to both ZEB1/2. Importantly, the ZEB1–Smad complex activates TGF-β signaling including transcriptional regulation and growth inhibition, while the ZEB2–Smad complex inhibits it. Phosphorylation of ZEB1 regulates its transcriptional activity via interaction with other transcriptional regulators. Phosphorylation at Thr851, Ser852, and Ser853 by PKC regulates IGF-1 signals, whereas ERK phosphorylates the consensus site of Thr867 in ZEB1 and allows hierarchical phosphorylation of Thr851/Ser852/Ser853 by PKC [42]. Consequently, these phosphorylation events inhibit ZEB1 activity. Ser585 of ZEB1 is phosphorylated by ataxia telangiectasia mutated (ATM) to promote its stabilization [43,44]. In ZEB2, phosphorylation of Ser705 and Tyr802 by GSK3β is reported to reduce its protein stability [45].

## 4. Synergy between TGF-β and Growth Factors in EMT

Epidermal growth factor (EGF), tumor necrosis factor-α (TNF-α), and Wnt signals are reported to cooperate with TGF-β signaling during cancer progression [3]. The synergy between TGF-β and RAS signaling has been thoroughly investigated through in vitro experiments as well as mice models. SNAIL, a representative EMT-TF, is induced by TGF-β in certain kinds of cancer cells, and is dramatically enhanced by active RAS signaling such as H-RAS G12V and K-RAS G12D [46,47]. Thus, cancer cells with the active K-RAS mutation, such as pancreatic carcinoma Panc-1, lung carcinoma A549, and prostate carcinoma PC3 cells, show constitutively activated RAS signaling, and exhibit drastic induction of SNAIL via TGF-β alone. TGF-β also synergistically induced SNAIL expression in HeLa cells expressing RAS G12V, whereas TGF-β failed to perform this function in HeLa cells in the absence of RAS mutations [46]. In addition to active RAS signaling, TGF-β in cooperation with growth factors such as EGF, fibroblast growth factor (FGF), and hepatocyte growth factor (HGF) synergistically induced SNAIL expression [46,47]. By contrast, the direct target genes of the TGF-β-Smad pathway, including Smad7 and plasminogen activator inhibitor 1, were not synergistically enhanced or marginally suppressed by active RAS signaling. The underlying molecular mechanisms are not fully understood. However, it is reported that signal transducer and activator of transcription 3 (STAT3) synergizes TGF-β and active RAS signals for SNAIL induction [48] (Figure 1). When STAT3 was silenced by its specific siRNAs, SNAIL induction by TGF-β and active RAS was dramatically inhibited. STAT3 has two major phosphorylation sites (Tyr705 and Ser727) which were phosphorylated by interleukin-6 (IL-6) family cytokines and active K-RAS, respectively. Phosphorylation of Tyr705 and Ser727 are indispensable and dispensable, respectively, for SNAIL induction by the cooperation of TGF-β with active RAS signaling. Recently, RAS-responsive element-binding protein 1 (RREB1), an RAS transcriptional effector, acted as a mediator of SNAIL induction by TGF-β in cooperation with active RAS in mouse models [47] (Figure 1). In this case, RREB1 activated by RAS-MEK-ERK collaborated with Smads activated by TGF-β receptors. Indeed, phosphorylation of RREB1 by ERK at Ser161 and Ser970 residues can interact with phosphorylated Smad to induce SNAIL expression [46,47]. Therefore, cancer cells with a highly activated ERK pathway exhibit a drastic induction of SNAIL by TGF-β alone, and active RAS and TGF-β cooperate to selectively induce SNAIL and EMT (Figure 1).

## 5. Unique Synergy between TGF-β and FGF through Regulation of Alternative Splicing during EMT

The underlying mechanism of enhancement of EMT by FGF2 and FGF4 is very unique. FGF receptor genes encode four functional receptors (FGFR1-FGFR4) that consist of an intracellular tyrosine kinase domain, a transmembrane domain, and three extracellular immunoglobulin-like domains (Ig-I, Ig-II, and Ig-III) [49]. Alternative splicing of the Ig-III domain produces either the IIIb (FGFR1IIIb-FGFR3IIIb) or IIIc (FGFR1IIIc-FGFR3IIIc) isoforms of the receptors, with distinctly different biological impacts based on their FGF binding specificities. The IIIb isoform responds to FGF-7, also known as KGF, and FGF-10, and is mainly expressed in epithelial cells. By contrast, the IIIc isoform responds to FGF-2 and FGF-4, and is expressed in stromal mesenchymal cells. Exposure of TGF-β in epithelial cells and cancer cells increased the expression of FGFR1 and decreased the expression of FGFR2 [50]. In addition, TGF-β changed the alternative splicing of FGFR from IIIb to IIIc by repressing the expression of epithelial splicing regulatory proteins 1 and 2 (ESRP1/2) [50,51] (Figure 3). In addition, both ZEB1 and ZEB2 interacted with the promoter region of ESRP1 and ESRP2, and repressed their expression. Thus, epithelial cells undergoing EMT and aggressive cancer cells become sensitive to FGF-2 and FGF-4, likely through upregulated FGFR1-IIIc expression. Therefore, FGF-2 and FGF-4, abundant in stromal tissues, enhance EMT of invasive cancer cells, and sustain aggressive phenotypes. Interestingly, cancer cells that become resistant to the anti-cancer drug trametinib, a MEK inhibitor, regain the ability to proliferate via reactivation through FGFR1 signals [52].

## 6. EMT Transcription Factors (EMT-TFs) in Breast Cancer and Head and Neck Squamous Carcinoma (HNSCC) Cells

Transcriptional repression of E-cadherin is found in various solid cancers and is sometimes mediated by EMT-TFs [53]. Among these, ZEB1/2 correlate positively with EMT phenotypes and the aggressiveness of breast cancer and HNSCC cells [51,54]. Breast cancer is classified into two subtypes, luminal and basal-like, and is well characterized in EMT studies [55,56]. The basal-like subtype is associated with poor prognosis and aggressive behavior and retains low levels of E-cadherin and high levels of ZEB1/2, while the luminal subtype shows the opposite expression profiles. Intriguingly, the SNAIL and Twist families are not correlated with the mesenchymal trait in breast cancer and HNSCC cells [51,54]. Thus, ZEB1/2 are positively correlated with EMT phenotypes and the aggressiveness of cancers, but SNAIL and Twist contribute to aggressiveness without mesenchymal phenotypes. As mentioned above, SNAIL expression is dependent on cooperation between active RAS and TGF-β signals in aggressive cancer cells. However, ZEB1/2 at high levels is sustained by ERK pathways [57,58]. In addition to the ERK pathway, ZEB1/2 could be regulated by NF-κB signaling pathways, as the basal-like subtype of breast cancer cells exhibits elevated NF-κB activity more than the luminal subtype [59]. Although it is unclear why NF-κB is constitutively activated in the basal-like subtype, blocking NF-κB/p65 activity by siRNAs and inhibitors resulted in a downregulation of ZEB1/2, which is accompanied by decreased aggressive phenotypes [60]. In fact, whether NF-κB/p65 directly binds to the promoter region of ZEB1/2 in cancer cells remains unclear. In colorectal cancer, aggressive cancer cells where both the ERK and NF-kB pathways are activated show high ZEB1 levels through FOXK2 transcription factors activated by both signaling pathways [60]. Therefore, sustained high expression of ZEB1/2 in cancer cells is highly dependent on the ERK and NF-κB pathways.

The pathological roles of SNAIL and ZEB1/2 are still controversial during cancer progression. In a mouse pancreas cancer model, cancer aggravation, including tumor differentiation, invasion, and metastasis, is not affected by genetic SNAIL depletion, but is reduced by genetic depletion of ZEB1 [61,62]. However, downregulation of ZEB1/2 through overexpression of miR-200, a known microRNA that directly targets ZEB1/2, in a mouse breast cancer model does not affect lung metastasis, but contributes to recurrent lung metastasis after chemotherapy [63]. Thus, EMT-TFs seem to have specific and complementary functions which are not redundant. Moreover, the roles of EMT-TFs could be tissue-specific, as demonstrated by the different roles of SNAIL and ZEB1/2 in distant metastasis of various cancers.

## 7. Pivotal Roles of Ets Family Proteins to Regulate SNAIL and ZEB in Cancer Cells

MEK–ERK inhibition downregulated SNAIL as well as ZEB1/2. The E26 transformation-specific (Ets) family of transcription factors has twenty-eight members in the human genome, and regulates many different biological and pathological processes [64,65]. Ets transcription factors have been identified as mediators of RAS/ERK signaling, and phosphorylation of Ets proteins by ERK can promote its transcriptional activity. Among the Ets family of transcription factors, Ets1, a prototypic member of this family, activates the *ZEB1* promoter and induces endogenous ZEB1/2 expression in cancer cells [57]. In addition, Ets1 activates the *SNAI1*-promoter and induces endogenous SNAIL expression [58]. Thus, Ets1 regulates the expression of both SNAIL and ZEB1/2, likely dependent on the MEK–ERK pathway (Figure 1). More importantly, epithelium-specific ETS (ESE) transcription factors are a subgroup of ETS transcription factors defined by shared homology of the ETS domain, and include ESE1 (ELF3, E74-Like Factor 3), ESE2 (ELF5, E74-Like Factor 5), and ESE3 (EHF, Ets homologous factor) [66]. ESE1 is expressed in many different organs, whereas expression of ESE2 and ESE3 is relatively restricted to glandular organs. Among these ESE proteins, ESE1 and ESE3 suppressed ZEB1/2 expression in cancer cells [54,67,68], whereas ESE2 failed to perform this function. Importantly, missense mutations in ESE1 and ESE3, which are found in cancer, abolish their function and promote EMT during cancer progression. Therefore, Ets family proteins define the EMT state through the regulation of the EMT-TFs.

## 8. ZEB Family Proteins Are Sustained by Dual Positive Feedback Loops

Aggressive cancer cells exhibit sustained high levels of ZEB1/2. ZEB1/2 directly bind to the promoter region of ESRP1/2 and transcriptionally suppress them, leading to regulating alternative splicing machinery of various mRNAs, including FGFRs, CD44, Mena (a member of the Enabled (Ena)/vasodilator-stimulated phosphoprotein family of proteins), and Rac1. As described above, when ZEB1/2 were upregulated and maintained at high levels in aggressive cancer cells, the IIIc isoforms of FGFRs were expressed through isoform switching when ESRP1/2 expression was repressed. Due to preferential binding of FGF2 and FGF4 to the IIIc-isoforms of FGFRs, the cancer cells expressing IIIc isoforms of FGFRs become sensitive to FGF2 and FGF4, which are abundant in stromal tissues. FGF2/4-bound FGFRs activate ERK pathways and maintain high levels of ZEB1/2 expression via activating ETS pathways, leading to a positive feedback loop of ERK–ZEB1/2–ESRP1/2–FGFRIIIc. Indeed, a positive correlation between ZEB1/2 and IIIc isoforms of FGFRs is observed in breast cancer cells, especially the basal-like subtype of breast cancer cells, such as BT549, MDA-MB-231, and Hs578T cells. These cells, with high ZEB1/2 levels, expressed only IIIc isoforms of FGFRs, with low ESRP1/2 levels [51] (Figure 4).

Hyaluronic acid, also called hyaluronan, is a glycosaminoglycan distributed widely throughout epithelial and stromal tissues. Hyaluronan is a major component of extracellular matrices, and has pivotal roles in cell proliferation, migration, and invasion during cancer progression, through its binding to cell surface receptors, CD44. It is reported that CD44 has an affinity towards other ligands such as fibronectin, collagen, chondroitin, and sulfated proteoglycan [69]. Isoform switching of CD44 mRNA is also regulated by ESRP1/2; ESRP1/2 activate the splicing of CD44 variants (CD44v) and silence that of the CD44 standard form (CD44s). Thus, when ESRP1/2 are downregulated by ZEB1/2, cancer cells express CD44s alone. Both CD44v and CD44s bind to hyaluronan, while CD44v have a higher binding affinity for hyaluronan than CD44s. Thus, it is reported that CD44v, specifically CD44v6, promote tumor progression and metastatic potential in lung, breast, and colon cancer [69]. Interestingly, the cells overexpressing CD44s show dramatic induction of cell-associated hyaluronan, and lower levels of hyaluronan in culture medium [70]. Since CD44 activates ERK pathways, ZEB1/2–ESRP1/2–CD44s/hyaluronan forms a positive feedback loop. Cancer cells with high levels of ZEB1/2 show low levels of ESRP1/2 and high levels of CD44s. Hyaluronan-bound CD44s provoke the ERK signaling pathway to maintain ZEB1/2 at high levels and increase the expression of hyaluronan (Figure 4).

## 9. Molecules That Regulate EMT as Potential Diagnostic Markers

Since EMT processes occur in normal embryogenesis, normal fibroblasts and some stromal cells constitutively express EMT-TFs at high levels. This makes it hard to use EMT-TFs as diagnostic markers that specifically detect cancer cells undergoing EMT or partial EMT. During EMT, a large number of mRNAs are changed by alternative splicing, producing divergent gene products [71]. Thus, this functional diversity seems to be an essential feature to define the epithelial, mesenchymal, or hybrid phenotypes [72]. Most importantly, oncogenic K-Ras contributes to EMT induction, suggesting that gene products generated from alternative splicing mediated by oncogenic K-Ras are limited in cancer cells, and differ from those produced during pathophysiological/fibrosis EMT and physiological/developmental EMT. Therefore, oncogenic K-Ras-mediated gene products would be useful as potential diagnostic molecular markers that specifically identify cancer cells undergoing EMT or partial EMT.

## 10. Conclusions and Perspectives

It has become increasingly clear that the expression of EMT-TFs is regulated transcriptionally by secreted factors and extracellular matrices, post-transcriptionally by several microRNAs, and post-translationally by ubiquitin-dependent degradation. Additionally, the function of EMT-TFs is controlled by various post-translational modifications. Consequently, EMT-TFs modify the expression of EMT marker proteins at the transcriptional and epigenetic levels. TGF-β autonomously secreted from cancer cells regulates the expression of EMT-TFs and acts as a cancer promoter in cooperation with active K-RAS mutants and other factors secreted from stromal cells in the cancer microenvironment. Thus far, it is still unclear to what extent cells undergo a conversion of cell morphology by partial EMT in vivo during cancer progression. In addition, recent studies have demonstrated that a partial EMT-like morphology or expression of EMT-TFs is associated with CTC and CSC. Therefore, understanding the molecules underlying partial EMT would be clinically important for both the diagnosis and treatment of cancer.

## Figures and Tables

**Figure 1 diagnostics-12-02127-f001:**
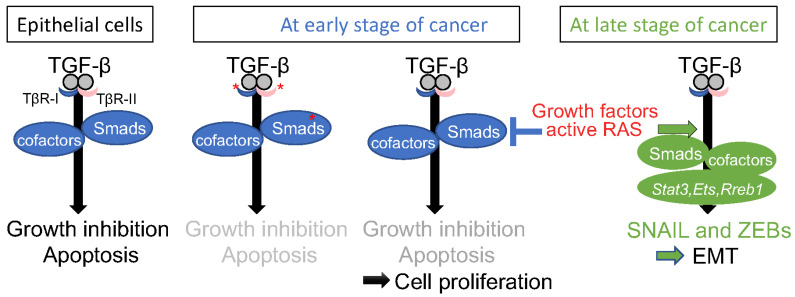
Synergism between TGF-β and RAS signals for EMT induction. In normal epithelial cells, TGF-β induces apoptosis and suppresses cellular proliferation of the cells (left). Once genetic alterations in signaling components of TGF-β occur, the cells become resistant to the cancer-suppressing effect of TGF-β (center). After RAS G12V mutation occurs even in the cells with wild-type of signaling components of TGF-β, it suppresses TGF-β-induced growth inhibition at early stages of cancer, but enhances TGF-β-induced SNAIL and EMT at late stages of cancer (right). * indicates mutations; genetic alterations in signaling components of TGF-β are infrequently observed in various types of cancers.

**Figure 2 diagnostics-12-02127-f002:**
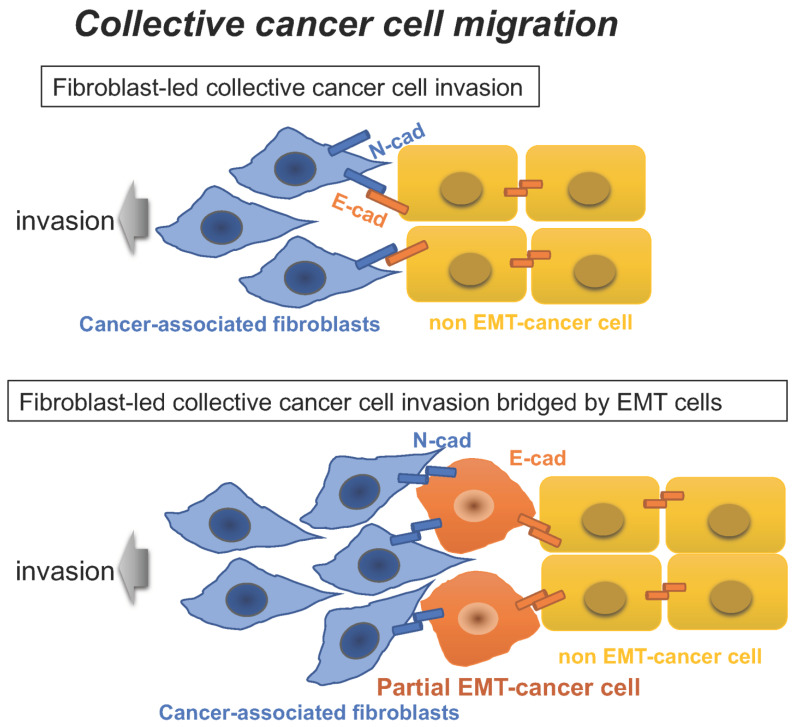
Collective cancer cell migration. Collective cancer cell migration is characterized by multicellular clusters analogous to a bunch of grapes, and includes fibroblast-led collective cancer cell migration/invasion (top) and promotion of invasion of non-EMT cancer cells through the interaction between non-EMT cancer cells and cancer-associated fibroblasts, which is bridged by EMT-cancer cells (bottom). N-cad, N-cadherin; E-cad, E-cadherin.

**Figure 3 diagnostics-12-02127-f003:**
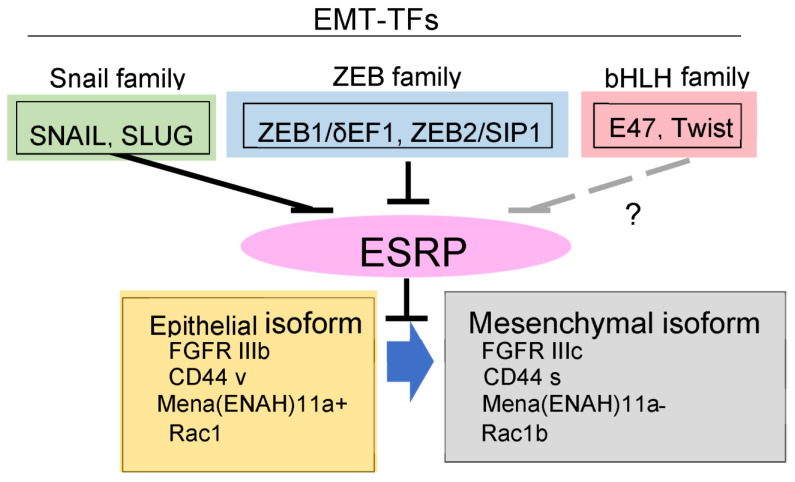
Schematic illustrations of EMT regulated by EMT transcription factors (EMT-TFs). Upregulation of EMT-TFs decreases expression of ESRPs, and subsequently changes in alternative splicing events during EMT. FGFR, fibroblast growth factor receptor; Mena (also known as ENAH), a member of the Enabled (Ena)/vasodilator-stimulated phosphoprotein (VASP) family of proteins; Rac1, RAS-Related C3 Botulinum Toxin Substrate 1.

**Figure 4 diagnostics-12-02127-f004:**
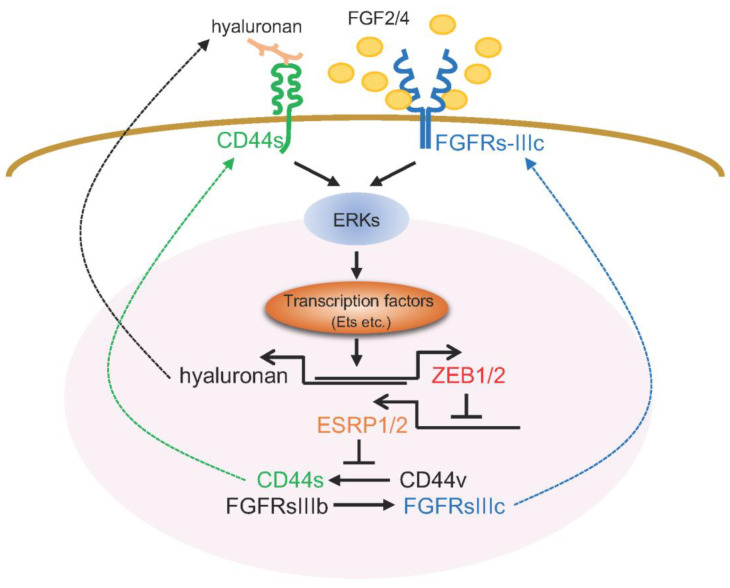
ZEB family proteins are sustained by dual positive feedback loops. Activated ERK1/2 pathways increase expression of both ZEB1/2 and hyaluronan. ZEB1/2 then downregulate ESRP1/2, resulting in expression of FGFRs-IIIc and CD44s isoforms. FGF2/4, abundant in stromal tissues, and hyaluronan bind FGFRsIIIc receptors and CD44s, respectively, and then provoke the ERK signaling pathway to maintain ZEB1/2 at high levels.

## Data Availability

Not applicable.

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
