# Peer review of "Epithelial–Mesenchymal Transition by Synergy between Transforming Growth Factor-β and Growth Factors in Cancer Progression"

_diagnostics, 2022, doi:10.3390/diagnostics12092127_

Round 1

Reviewer 1 Report

Manuscript ID: Diagnostics 1777158

Type of manuscript: Review

Journal: Diagnostics

Title: Epithelial-Mesenchymal Transition by Synergy Between Transforming Growth Factor-β and Growth Factors in Cancer Progression

The author, in this review, describes to the importance of biological and pathological function of EMT and TGF beta in the tumor progression.

The paper is well organized and well written but does not represent a contribution to the field of the EMT and TGF beta. I personally recommend the publication of this work with major revision.

I suggest add new information to the manuscript with updated literature to make the review more interesting. Indeed, the same author wrote another similar review in 2015:”Epithelial-mesenchymal transition is regulated at post-transcriptional levels by transforming growth factor-β signaling during tumor progression. Saitoh M. Cancer Sci. 2015 May;106(5):481-8”.

Therefore, it would be desirable to add some images relating to the mechanisms described.

Author Response

I suggest add new information to the manuscript with updated literature to make the review more interesting. Indeed, the same author wrote another similar review in 2015:”Epithelial-mesenchymal transition is regulated at post-transcriptional levels by transforming growth factor-β signaling during tumor progression. Saitoh M. Cancer Sci. 2015 May;106(5):481-8”. Therefore, it would be desirable to add some images relating to the mechanisms described.

A; First of all, I would like to express my deepest gratitude for your kind assessment of my revision.Thank you very much for valuable comments. I agree with the comment from the reviewer 1, but I focused on basic mechanisms, rather than the latest molecular mechanism, of EMT by TGF-beta in this manuscript likely due to aim and scope of “Diagnostics” journal. I updated references and added one more figure (Figure 4) describing the mechanisms of dual positive feedback loops of ZEB.

Reviewer 2 Report

Title

Epithelial-Mesenchymal Transition by Synergy Between Transforming Growth Factor-β and Growth Factors in Cancer Progression.

Concise Summary

The author aims to review epithelial-mesenchymal transition (EMT) in cancer progression, because EMT is associated with all the activities of tumor cells. EMT is induced by transcription factors which are regulated by growth factors, cytokines and extracellular matrix components. The author state that transforming growth factor (TGF-β) acts as the key mediator for EMT during cancer progression. It is concluded that understanding the molecules underlying partial EMT is clinically important for both the diagnosis and treatment of cancer.

Major Comments

The article is generally well written and the considerations about the issue are adequate and interesting. However, there are a number of considerations to improve the consistency of the article:

1. Conclusions & Perspectives. This section deals with topics that have not been discussed previously, as for example:

a. It is said that “it has become increasingly clear that the expression of the EMT-TFs is regulated transcriptionally by secreted factors, extracellular matrices, and exosomes.” However, exosomes are not mentioned in any part of the text.

b. The author comments that “So far, it is still unclear to what extent cells undergo a conversion of cell morphology by partial EMT in vivo during cancer progression”. However, in the text partial EMT in vivo cancer progression is not discussed at all.

c. The author states that “ it is not well understood whether EMT contributes to metastasis of sarcoma cells.” However, nothing is discussed in the article about the role of EMT in the sarcoma metastasis process.

d. The author concludes that “… understanding the molecules underlying partial EMT is clinically important for both the diagnosis and treatment of cancer.” However, the clinical relevance of partial EMT in the diagnosis and treatment of cancer is not examined in the article”.

e. The conclusion: “Therefore, in future studies based on molecular pathology and cell biology, pharmacological targets for the partial EMT machinery remain to be elucidated” is confusing and is non-informative.

f. In addition, in this section it is said that “recent studies have demonstrated that a partial EMT-like morphology or expression of EMT-TFs is associated with CTC and CSC.”  However, the information given in the text about partial EMT-like morphology is limited.  In addition, only two references have been included in the bibliography about this issue, and these referred articles are dated in 2018. Could you explain why there are not more recent references about this topic?

2. References. There are only ten out of 68 references from 2019 till 2022. It is supposed that should have more recent references about this active research topic. Could you explain the reason to discuss only a few references?

3. Unfortunately, figures have not enough quality and should be improved.

Conclusion

Finally, I consider that it is an interesting review that gives relevant information about the role of TGF-β in EMT process and oncogenesis. The author considerations about the issue of the article are adequate, although insufficient. I suggest improving the quality of the text and figures in the manuscript.

Author Response

First of all, I would like to express my deepest gratitude for your kind assessment of my revision.

1. Conclusions & Perspectives. This section deals with topics that have not been discussed previously, as for example:

2. It is said that “it has become increasingly clear that the expression of the EMT-TFs is regulated transcriptionally by secreted factors, extracellular matrices, and exosomes.” However, exosomes are not mentioned in any part of the text.

A: Exosome has very complicated functions during cancer progression and EMT. To avoid confusion of readers, the term of “exosome” is removed in the main text.

3. The author comments that “So far, it is still unclear to what extent cells undergo a conversion of cell morphology by partial EMT in vivo during cancer progression”. However, in the text partial EMT in vivo cancer progression is not discussed at all.

A: Concerning “partial EMT”, I mentioned it in third paragraph of section “Collective or single cancer cell migration by EMT”

4. The author states that “ it is not well understood whether EMT contributes to metastasis of sarcoma cells.” However, nothing is discussed in the article about the role of EMT in the sarcoma metastasis process.

A: Several papers on EMT in sarcoma cells have been published, but most of them described an involvement of miRNAs of the cells. Indeed, I and my colleagues are now working on EMT in sarcoma in vitro and in vivo, and found that it is not easy to evaluate the roles of the EMT-TFs in sarcoma cells, because sarcoma cells exhibited high expression of almost all EMT-TFs. To avoid confusion of readers, I removed the term of “sarcoma” and modified the last part of manuscript.

5. The author concludes that “… understanding the molecules underlying partial EMT is clinically important for both the diagnosis and treatment of cancer.” However, the clinical relevance of partial EMT in the diagnosis and treatment of cancer is not examined in the article”.

A: Thank you very much. I modify this sentence.

6. The conclusion: “Therefore, in future studies based on molecular pathology and cell biology, pharmacological targets for the partial EMT machinery remain to be elucidated” is confusing and is non-informative.

A: These sentences are deleted and modified in the main text.

7. In addition, in this section it is said that “recent studies have demonstrated that a partial EMT-like morphology or expression of EMT-TFs is associated with CTC and CSC.” However, the information given in the text about partial EMT-like morphology is limited. In addition, only two references have been included in the bibliography about this issue, and these referred articles are dated in 2018. Could you explain why there are not more recent references about this topic?

A: I (we) know that a lot of papers on partial or hybrid EMT are published, but unfortunately I and my colleagues have failed to reproduce the results of some papers. Indeed, although we read thoroughly recent many papers on partial EMT or hybrid EMT, I (we) have sometimes serious concerns with their experiments, results, and conclusion. Thus, I hesitate to cite these papers in this manuscript. However, I agree with the reviewer’s comments, and updated the reference with addition of some references.

8. References. There are only ten out of 68 references from 2019 till 2022. It is supposed that should have more recent references about this active research topic. Could you explain the reason to discuss only a few references?

A: I agree with the reviewer’s comments, and updated the reference with addition of some references.

9. Unfortunately, figures have not enough quality and should be improved.

A: The image is now improved, and new figure 4 is added in the revised manuscript.

Round 2

Reviewer 1 Report

I recommend it in this present form.